# Evaluation of the Mitragynine Content, Levels of Toxic Metals and the Presence of Microbes in Kratom Products Purchased in the Western Suburbs of Chicago

**DOI:** 10.3390/ijerph17155512

**Published:** 2020-07-30

**Authors:** Walter C. Prozialeck, Joshua R. Edwards, Peter C. Lamar, Balbina J. Plotkin, Ira M. Sigar, Oliver Grundmann, Charles A. Veltri

**Affiliations:** 1Department of Pharmacology, College of Graduate Studies, Midwestern University, 555 31st Street, Downers Grove, IL 60515, USA; jedwar@midwestern.edu (J.R.E.); plamar@midwestern.edu (P.C.L.); 2Department of Microbiology and Immunology, College of Graduate Studies, Midwestern University, 555 31st Street, Downers Grove, IL 60515, USA; bplotk@midwestern.edu (B.J.P.); isigar@midwestern.edu (I.M.S.); 3Department of Pharmaceutical Sciences, College of Pharmacy Glendale, Midwestern University, 19555 N. 59th Avenue, Glendale, AZ 85308, USA; ogrund@midwestern.edu (O.G.); cveltr@midwestern.edu (C.A.V.)

**Keywords:** kratom, microbes, Salmonella, metals, Pb, Ni, Cr

## Abstract

Kratom (*Mitragyna speciosa*, Korth) is a tree-like plant that is indigenous to Southeast Asia. Kratom leaf products have been used in traditional folk medicine for their unique combination of stimulant and opioid-like effects. Kratom is being increasingly used in the West for its reputed benefits in the treatment of pain, depression and opioid use disorder. Recently, the United States Food and Drug Administration and Centers for Disease Control have raised concerns regarding the contamination of some kratom products with toxic metals (Pb and Ni) and microbes such as *Salmonella*. To further explore this issue, eight different kratom products were legally purchased from various “head”/”smoke” shops in the Western Suburbs of Chicago and then tested for microbial burden, a panel of metals (Ni, Pb, Cr, As, Hg, Cd), and levels of the main psychoactive alkaloid mitragynine. All of the samples contained significant, but variable, levels of mitragynine (3.9–62.1 mg/g), indicating that the products were, in fact, derived from kratom. All but two of the samples tested positive for the presence of various microbes including bacteria and fungi. However, none of the samples tested positive for *Salmonella*. Seven products showed significant levels of Ni (0.73–7.4 µg/g), Pb (0.16–1.6 µg/g) and Cr (0.21–5.7 µg/g) while the other product was negative for metals. These data indicate that many kratom products contain variable levels of mitragynine and can contain significant levels of toxic metals and microbes. These findings highlight the need for more stringent standards for the production and sale of kratom products.

## 1. Introduction

Kratom (also known as ketum) is a tree-like plant (*Mitragyna speciosa*, Korth, Rubiaceae) that is native to Thailand, Malaysia, Indonesia and other regions of Southeast Asia [1,2,3]. For generations, indigenous peoples in Southeast Asia have used kratom leaves (either unprocessed or brewed into teas or other decoctions) as a mild stimulant to stave off fatigue, or as an opioid substitute to treat pain or opioid use disorder [4,5,6]. Pharmacologic studies have shown that kratom leaves contain a variety of active alkaloids with the most significant being mitragynine and 7-hydroxymitragynine [7,8]. Mitragynine has partial agonist activity at mu-type opioid receptors and antagonist activity at delta-type opioid receptors [7,8,9,10]. In addition, it can modulate the release and synaptic actions of neurotransmitters such as norepinephrine and serotonin [7,10,11]. Even though kratom has been used in Southeast Asia for generations, it is only over the past 25 years that kratom use has expanded to Europe and North America [1,2,4,12,13,14]. In the US, kratom products are used extensively for the self-management of pain, depression and opioid use disorders [15,16,17,18]. The most widely used products include chopped, dried leaf material (either alone or in capsule or tablet form) or concentrated extracts that are formulated as liquids or capsules [2,12,19]. These products are widely available from internet vendors or in specialty stores commonly known as “head shops” or “smoke shops”, although some products are now being sold through chain stores that specialize in the sale of herbal supplements.

In the US, kratom is regarded as a new dietary ingredient under the United States Food and Drug Administration (FDA) and Drug Enforcement Administration policies, and is not regulated as a dietary supplement according to the Dietary Supplement Health and Education Act (for reviews see: [12,19,20]). Although it remains legal in most of the US, at the time of writing several states, such as Alabama, Florida, Indiana, Arkansas, Wisconsin and Tennessee, have passed legislation banning the local sale and possession of kratom (for reviews see [19]).

As kratom use in the US has increased, there has been growing concern about the safety of kratom and kratom products being sold in the US. Much of the concern focused on the increase in the number of reports to poison control centers in which kratom was mentioned as a contributing factor in adverse health effects including up to 90 deaths [21,22,23]. These reports of adverse effects have been cited as the basis for the Drug Enforcement Agency (DEA)’s proposal to place kratom and its mitragynine alkaloids into Schedule I of the Controlled Substance Act [24], which would effectively ban the sale and possession of kratom. However, there was no definitive proof that kratom was the cause of the deaths. Almost all cases involved the use of other drugs including opioids, benzodiazepines or alcohol, or the presence of serious health conditions including refractory depression [25,26,27,28]. In some cases, the kratom products themselves may have been adulterated with exogenous materials such as synthetic opioids [25,29]. The DEA’s proposal to ban kratom prompted a massive response by kratom users and advocates that led the DEA to place the move to Schedule I status on an indefinite hold that continues to the present day [19].

Recently, additional safety concerns have been raised about kratom products in the US. Two of the most important concerns involve the possible contamination of kratom products with harmful microbes, particularly *Salmonella* [30,31,32], and toxic metals, such as Ni and Pb [33,34]. In order to further investigate these contamination issues, we evaluated the levels of microbes, toxic metals (As, Cd, Cr, Fe, Hg, Ni, and Pb), mitragynine, and possible adulteration with benzodiazepines and opioids in a panel of eight different kratom products that were purchased in the western suburbs of Chicago.

## 2. Materials and Methods

### 2.1. Source of Products

The kratom products analyzed in the present study were legally purchased from “head” shops and “smoke” shops in several western suburbs of Chicago, IL (Westmont, IL, Lisle, IL, Elmhurst, IL and Aurora, IL, USA) in July and August of 2019. The samples were stored in the dark, at room temperature, in their original packaging, until processing for the various analyses. To ensure consistency of results, 2 samples of each product, which were purchased about 1 month apart, were evaluated. The samples were initially opened under sterile conditions in a laminar flow hood and portions were processed for microbial culture. The remaining portions of each sample were aseptically divided and processed for the analysis of a panel of metals, mitragynine content, and screening for 13 opioids and 8 benzodiazepines.

### 2.2. Microbiological Testing

To determine the prevalence of *Salmonella* contamination associated with the kratom samples obtained, each sample (1 or 0.5 gm) was suspended in sterile phosphate buffered saline (PBS) (1:10 dilution) and vortexed (1 min). The sample was then serially diluted and plated (*n* = 3) onto sheep blood agar, an enriched, differential (hemolysis) bacteriological medium used to permit the broadest range of bacteria to grow, and MacConkey agar, a medium selective for Gram negative bacteria and differential for lactose fermentation, which is consistent with bacteria that are members of the order Enterobacteriales. All lactose negative colonies were screened for oxidase production to validate *Salmonella* testing (oxidase negative) and MacConkey agar by standard spread plate method. Plates were incubated at 37 °C and 42 °C for 24–48 h. After 24 and 48 h incubation, the number of colonies on each medium was counted. Screening for the presence of *Salmonella* was done using the methods outlined in the FDA *Official Methods for Bacteriological Analytical Manual* [35]. Briefly, kratom samples from each sample (*n* = 16) were enriched for *Salmonella* in Neogen Revive/RV-M broth per the manufacturers’ directions (Reveal 2.0 for *Salmonella*, Neogen, Lansing, MI, USA). Portions of the enriched samples that were positive with Reveal 2.0 *Salmonella* were then spread plated onto both MacConkey and Hektoen Enteric (HE) agar. All lactose negative colonies were screened for oxidase production. Verification of the identity of lactose-negative colonies was performed using an API-20 test kit (BioMerieux, Durham, UK) per the manufacturer’s instructions. All suspected *Salmonella* colonies from HE were inoculated into Triple Sugar Iron agar and lysine iron agar slants, per FDA protocol [35]. All tests were negative for *Salmonella*.

### 2.3. Mitragynine Analysis

The samples (1 g) were placed in individual teabags and extracted with 12 mL of 100% methanol (ACS Grade, Fisher Scientific, Waltham, MA, USA) using microwave assisted extraction (MAE) with an Ethos EX labstation (Milestone Srl, Sorisole, Bergamo, Italy). The extraction conditions were a 15 min ramp to 65 °C, held at 65 °C for 30 min, followed by a 15 min cool-down, for a total run time of 4 °C.

For evaluation of mitragynine and 7-hydroxymitragynine concentrations and detection of benzodiazepines, extracts were diluted to 25 mg/mL, 1 mg/mL, and 0.1 mg/mL and placed in the autosampler rack of an Agilent 1260 HPLC system (Agilent Technologies, Santa Clara, CA, USA). For each HPLC separation, 1 µL was injected and chromatographed on an Agilent Poroshell 120 EC-C18 column (4.6 × 100 mm, 2.7 µm, Agilent Technologies, Santa Clara, CA, USA), equipped with the appropriate guard column, and separated at a flow rate of 0.5 mL/min. The mobile phase consisted of eluent A (LCMS grade water with 0.1% formic acid, Fisher Scientific, Waltham, MA, USA) and eluent B (LCMS grade acetonitrile with 0.1% formic acid, Fisher Scientific, Waltham, MA, USA). An isocratic elution of 25% B was started for 1 min, followed by a linear gradient applied from 25% to 75% B over 11 min, finally an isocratic wash of 75% B for 5 min was applied before a re-equilibration of 25% B for 3 min. Typically, a back pressure of <95 bar was observed at 25% acetonitrile/water.

For detection of opioids, extracts (25 mg/mL) were placed in the autosampler rack of the HPLC system. For each HPLC separation, 1 µL was injected and chromatographed on an Agilent Poroshell 120 Phenyl-hexyl column (4.6 × 100 mm, 2.7 µm, Agilent Technologies, Santa Clara, CA, USA), equipped with the appropriate guard column, and separated at a flow rate of 1.44 mL/min. The mobile phase consisted of eluent A (LCMS grade water with 0.1% formic acid, Fisher Scientific) and eluent B (LCMS grade acetonitrile with 0.1% formic acid, Fisher Scientific). An isocratic elution of 10% B was started for 1 min, followed by a linear gradient applied from 10% to 45% B over 3 min, and then from 45% to 100% B over 2 min, finally an isocratic wash of 100% B for 2.4 min was applied before a re-equilibration of 10% B for 4 min. Typically, a back pressure of <95 bar was observed at 10% acetonitrile/water.

Compounds were detected using an Accurate-Mass 6530 quadrupole time-of-flight mass analyzer (Q-TOF, Agilent Technologies, Santa Clara, CA, USA). The Q-TOF mass analyzer provided identification of compounds using accurate masses of full spectra in Targeted MS/MS mode.

Quantification of mitragynine and 7-hydroxymitragynine were performed using a series of calibration controls prepared fresh on each day of analysis at incremental concentrations ranging from 1 ng/mL to 100 µg/mL. The coefficient of determination was >0.99 for all calibration curves. The extracted kratom samples were analyzed for the presence of 13 opioids and 8 benzodiazepines by comparing the chromatograms to those of the reference mixtures Pain Management Multi-component Opiate Mixture-13 solution and Benzodiazepine Multi-component Mixture-8 solution (Sigma Aldrich, St. Louis, MO, USA). All samples were analyzed using MassHunter Software (Agilent Technologies, Santa Clara, CA, USA).

### 2.4. Metal Analyses

Samples of each product were analyzed for a panel of metals (As, Cd, Cr, Fe, Hg, Ni and Pb) by S.G.S. Chemical Solutions, Inc. (Harrisburg, PA, USA). The contents of the capsule samples were emptied. All samples were weighed and dried to a constant weight prior to digestion. Approximately 0.5 g of sample was analytically weighed into a microwave digestion vessel. Five mL of concentrated nitric acid and 4 mL of 30% hydrogen peroxide were added to each sample vessel. The samples were digested in a Milestone Ethos microwave digester at a final temperature of 180 °C for 20 min after being ramped to that temperature in 20 min. The sample vessels were then removed, and the contents were transferred to virgin polypropylene 50 mL flat bottom tubes. They were further diluted in these tubes to a final volume of 25 mL with American Society for Testing Materials (ASTM) Type I deionized water. The sample digestates were then analyzed using Perkin Elmer Inductively coupled plasma-mass spectrometer (ICP-MS) instruments (Fisher Scientific, Waltham, MA, USA). Different external analytical calibration ranges were established depending on the element(s) of interest. Appropriate internal standards were added to each sample aliquot prior to further dilution and subsequent analysis. Results were expressed as µg/g dry weight and µg/g wet weight.

## 3. Results

Table 1 lists the eight kratom products that were evaluated. Six of the products contained finely ground leaf material (400–600 mg) in capsules (1, 1A, 2, 2A, 4, 4A, 5, 5A, 6, 6A) or as the free powder (8, 8A). Samples 3 and 3A contained processed leaf material in capsules and samples 7 and 7A contained a concentrated alkaloid extract in capsules. These products were chosen for analysis because they appeared to be widely available and were prominently displayed in many of the shops that were visited by the lead author. In addition, informal discussions with kratom users indicated that these products were being widely used in the Chicago area.

The results of the microbial analyses are summarized in Table 2. The overall levels of microbes associated with kratom ranged from zero to in excess of a million colony forming units (CFU) on the most permissive medium (sheep blood agar) incubated at human body temperature. There was also variability in the microbial load between different samples of product obtained from the same source. Interestingly, the most processed kratom products, samples 3 and 7, exhibited the lowest level of microbial contamination (<50 CFU/mL level of sensitivity). Bacteria identified as oxidase negative, lactose negative that were isolated from samples 4 and 5 were definitively identified as *Acinetobacter baumanni*/*calcoaceticus*, *Erwinia* spp., *Serratia* spp., *Pseudomonas oryzihabitans*, *Enterobacter* spp. or *Aeromonas hydrophila*/*caviae*/*sobria* via the API20 test system. None of the samples tested positive for the presence of *Salmonella*.

Table 3 lists the analytical parameters that were used for the analyses of mitragynine analogs and possible adulteration with a panel of opioids and benzodiazepines. Figure 1 shows representative chromatograms that were used to identify and quantify mitragynine. Table 4 summarizes the results of the mitragynine quantitation and adulteration analyses. Mitragynine was present in all samples indicating that the samples were derived from kratom. Samples 1–6 and 8 contained levels of mitragynine comparable to other commercial kratom products ranging from 3.35–11.33 mg mitragynine/g raw material while the content of mitragynine in sample 7 was much higher at 59.76 mg/g. None of the tested kratom products had detectable quantities sufficient for a positive presence of the 7-hydroxymitragynine, reference opioids and benzodiazepines.

Table 5 summarizes the results of the metal analyses in terms of µg of metal per gram of main product. Note that the values for the replicate samples are generally in excellent agreement with each other. All of the samples except #7 and 7A, which were concentrated kratom extracts, contained measurable levels of metals. As would be expected for leaf-based products, levels of Fe were consistently high and were similar to levels reported by Braley and Hondrogiannis [36]. Of the toxic metals that were evaluated, the highest levels were for Pb, Ni and Cr. Levels of several other metals including As, Cd and Hg were very low, actually at or near the level of detection.

It should be noted that, in evaluating the metal content of the samples, we primarily considered the results on the basis of the µg of the raw wet weight of each kratom product. This was done because kratom users typically base their kratom doses on grams of leaf product as obtained from vendors. Linking the metal levels to raw weight of product allows for direct estimation of the levels of metal that would occur with commonly used doses of kratom (See Discussion). The data for levels of metals based on dry weight of product (not shown) were consistently 10–15% higher than the results based on raw, wet weight. This difference was the same across all products tested.

## 4. Discussion

Of the eight kratom products that were evaluated in the present study, six were finely ground raw leaf products, either alone or in capsule form. Two of the products were concentrated (7, 7A) or semi-concentrated (3, 3A) kratom leaf extracts in capsule form. The three key findings were: (a) all of the samples contained substantial, but variable, levels of mitragynine, the main active constituent of kratom, while none of them were adulterated with detectable levels of 7-hydroxymitragynine, opioids or benzodiazepines; (b) all of the raw leaf products, but neither of the concentrated extracts, contained significant levels of microbial contamination; and (c) all of the raw leaf products contained potentially dangerous levels of toxic metals. Each of these findings has major ramifications for public health.

The fact that all of the samples contained significant levels of mitragynine is especially important. First, the levels of mitragynine that we found are comparable to those reported by other studies [37]. Moreover, mitragynine is a complex chemical that is difficult to synthesize [38]. Therefore, the presence of mitragynine in the samples strongly indicates that all of the products were derived from kratom plant material. In conjunction with the mitragynine studies, we also tried to measure levels of 7-hydroxymitragynine, which is an active constituent or metabolite of mitragynine. In addition, there is evidence that some unscrupulous purveyors of kratom may have fortified their products with exogenous 7-hydroxymitragynine [39]. The results of our studies indicated that all of the samples had very low or undetectable levels of 7-hydroxymitragynine (data not shown), which strongly suggests that the products were not fortified with exogenous 7-hydroxymitragynine. In conjunction with these studies, we also screened the samples for possible adulteration with opioids and benzodiazepines. Results of our studies found no evidence that any of these drugs were present in the samples. Adulteration with opioids such as oxycodone or fentanyl could increase the analgesic potency and dependency liability of kratom, creating a potential use disorder that would not occur with native kratom material alone [25,27,28,29]. Similarly, addition of an illicit benzodiazepine can enhance both the sedative and anxiolytic effects of kratom. A majority of kratom users self-treat conditions such as acute or chronic pain as well as mental or emotional disorders with the product [13,17]. Adulteration with prescription drugs would increase the potency and risk of dependence, leading to abuse.

Our finding that all of the ground-leaf preparations of kratom contained significant levels of microbes is not surprising. One would expect that any products derived from leaves of plants, under non-sterile conditions, would contain microbes. The two products that were processed extracts (3, 3A and 7, 7A) did not exhibit evidence of microbial contamination to the level of assay sensitivity. While we do not know how these samples were processed, it seems likely that the extraction procedures would involve the use of organic solvents and extreme acidic and alkaline pH’s, which would have antimicrobial activity. Previously, there have been reports of the presence of *Salmonella*, a pathogen associated with kratom products [31]. In contrast to that report, no *Salmonella* was isolated from any of the eight products tested.

A significant concern that has not been adequately addressed is the presence of *Acinetobacter* associated with loose leaf material. The presence of this intrinsically multidrug resistant opportunistic pathogen in easily aerosolized plant material could pose a threat to individuals with compromised immune systems who inhale the plant material [40]. Therefore, precautions should be exercised if using or dispensing loose-leaf kratom material.

One of the most important findings from this study is that all of the ground-leaf kratom products contained significant levels of toxic metals, particularly Ni and Pb. The United States FDA was the first agency [34] to raise concerns about the contamination of kratom products with potentially toxic levels of Ni and Pb, even though the study made no mention of other toxic metals. The results of the present analyses showed that seven of eight examined commercial kratom products contained relatively high levels of four metals including Fe, Pb, Ni and Cr, whereas they contained only trace levels of several more hazardous metals including As, Cd and Hg.

A critical question, of course, is, do any of these metals represent a hazard to kratom users? In considering this issue it is important to note that it is not uncommon for heavy kratom users to consume doses of 5–15 g of raw leaf material per day [12]. With this fact in mind, we believe that of the metals detected, the most problematic is Pb, which is classified as a Class I contaminant, indicating a high potential for toxicity, especially with chronic use [41]. Pb can cause serious neurological, psychological, cognitive, reproductive, developmental, immunologic, cardiovascular and renal effects [42]. The toxic neurological effects may be especially serious in children and young adults [42]. The permitted daily oral exposure level for Pb in foodstuffs and pharmaceuticals is only 5 µg/day [41]. The levels of Pb in several of the samples we tested were in the range of 0.25–1.6 µg/g product. This is significant because kratom users who commonly report consuming 5–15 g of kratom leaf per day could easily exceed the allowable daily intake of Pb (5 µg/day). This raises the possibility that some of the unusual toxicities of “kratom” products in the West may, at least in some cases, be partly attributable to Pb contamination. Further forensic studies are needed to clarify this issue.

The presence of relatively high levels of Ni (>2 µg/g) in most products could also represent a potential hazard to kratom users. Ni is classified as a Class 2A hazard, indicating that it is relatively abundant and has potential for causing serious toxicities [41]. Most significantly, Ni is classified as a Group I carcinogen (International Agency for Research on Cancer, IARC, 2012) and immunotoxin [41]. Despite these hazards of nickel, the daily allowable oral intake is set at 220 µg/day [41]. With the levels of Ni in the products tested in the present studies (>7.4 µg/g product), it is unlikely that even consumers who use high doses of kratom of >15 g raw leaf product per day would exceed allowable intake levels. On the other hand, it is unclear how chronic exposure to even these levels of Ni might affect human health; especially when ingested with the toxic levels of Pb in many of the samples.

Even though the levels of Cr in the leaf samples appeared to be quite high (up to 5.7 µg/g) it is unlikely that this represents a hazard to humans. Cr is classified as a Class 3 metal, with low potential for toxicity [41]. The permitted daily exposure level for Cr is 11,000 µg/day, which is a level of exposure that could not be achieved with any of the products even at kratom doses of over 15 g/day.

Levels of other metals (As, Cd and Hg) were well below allowable daily intake levels [41] and probably do not by themselves represent a significant hazard to kratom users. On the other hand, it is well established that exposure to mixtures of metals may increase risk of toxicity. Further studies are needed to determine if this is an issue with regard to kratom.

An obvious question that arises is, what is the source of the Ni and Pb in the raw-leaf samples? While we can only speculate at this time, one possibility is that the metals may break off or leach from the equipment that is used to grind, process, transport and store the kratom leaf material [41]. A second possibility is that the metals might be absorbed from the soil in which kratom is grown [41]. Ni is, in fact, an important nutrient for plants. It is noteworthy that most of the kratom sold in the United States is imported from sources in Indonesia [2,43,44]. The volcanic soil in many parts of Indonesia are known to contain high levels of metals, particularly Ni [45]. In addition, Pb pollution is a widespread problem in many regions of Indonesia [46,47]. Again, it is noteworthy that the samples that did not contain metals were concentrated extracts. It seems that the extraction procedure removed the metals. Further studies to examine the metal content of kratom samples from different geographic regions certainly seem warranted.

One limitation of the present study involves the relatively small number of products evaluated. It is unclear whether or not our finding of significant metal and microbial contamination applies to other kratom products. The American Kratom Association (AKA), a major kratom trade organization, has developed a set of “Good Manufacturing Practices (GMP)” for the kratom industry [48]. It is noteworthy that only one of the producers of the kratom products that were tested in the present study (O.P.M.S.) appears to have agreed to follow the American Kratom Association guidelines [49]. Interestingly, their products showed little or no evidence of contamination with microbes or metals (7, 7A).

## 5. Conclusions

We have advocated for further research on the therapeutic potential of kratom [19], and we stand by that position. However, we also find the present findings to be troubling. It is apparent that many of the kratom products being sold on the local level contain unknown levels of active agent (mitragynine) and are contaminated with metals, such as Pb and Ni, as well as microbes. This puts consumers at potential risk of adverse effects. Even though the AKA has adopted and advocated for GMP, it is obvious that many purveyors of kratom products have not adopted or adhered to those standards.

It is our hope that the present study will serve as a template for more extensive studies on the large number of kratom samples that are being sold in shops as well as through internet vendors. Such data are critical in formulating rational standards for the sale and production of kratom products.

## Figures and Tables

**Figure 1 ijerph-17-05512-f001:**
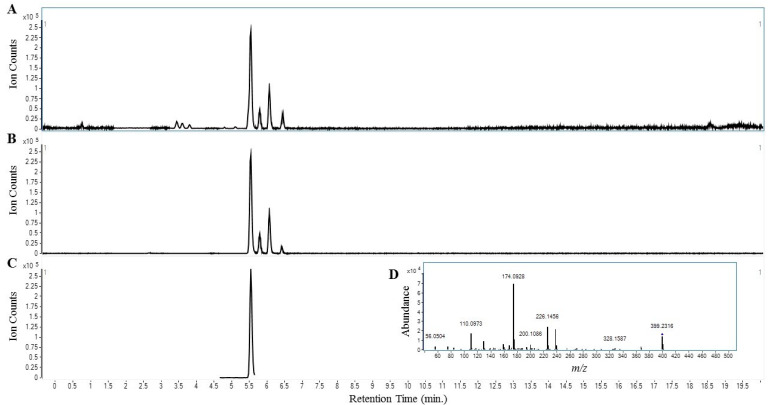
Representative Chromatograms used to Identify and Quantify Mitragynine. The kratom extract (100 µg/mL in methanol) was prepared and assayed as described in the methods section. (**A**) shows base peak chromatogram. (**B**) shows extracted ion chromatogram identifying chemistry with molecular formula C23H30N2O4 for mitragynine and related isomers. (**C**) shows extracted ion chromatogram identifying mitragynine using Targeted MS/MS transition of 399.23 to 174.09 *m*/*z* (**D**).

**Table 1 ijerph-17-05512-t001:** List of Kratom Products from Suburban Chicago.

Sample #	Product	Vendor
1 1A	Expert Botanicals Maeng Da Capsules	Westmont, IL, USA
2 2A	mK Botanicals Maeng Da Capsules	Westmont, IL, USA
3 3A	OPMS Maeng Da Capsules	Lisle, IL, USA
4 4A	CBD Kratom Thai Maeng Da Capsules	Elmhurst, IL, USA
5 5A	CBD Kratom Mixed Malay Capsules	Elmhurst, IL, USA
6 6A	CBD Kratom Indo Red Bantuagle Capsules	Elmhurst, IL, USA
7 7A	OPMS Gold Kratom Extract Capsules	Aurora, IL, USA
8 8A	NJOY Kratom Red Malay Powder	Lisle, IL, USA

**Table 2 ijerph-17-05512-t002:** Evaluation of kratom sample microbial load.

	CFU/g Product		
Product #	Sheep Blood Agar ^a^	MacConkey Agar ^b, c^	
	37 °C	37 °C	42 °C
1	>1 × 10^8^	6.7 × 10^4^	2.6 × 10^4^
1A	3.4 × 10^5^	6.5 × 10^4^	3 × 10^4^
2	3.3 × 10^5^	0	0
2A	2.5 × 10^5^	1	2
3	0	0	0
3A	0	0	0
4	>1 × 10^8^	4.1 × 10^5^	1.6 × 10^6^
4A	6.4 × 10^5^	3.2 × 10^5^	3.6 × 10^5^
5	>1 × 10^8^	8.7 × 10^4^	0
5A	2.9 × 10^5^	1.7 × 10^4^	2.0 × 10^4^
6	>1 × 10^8^	4.2 × 10^5^	0
6A	8.0 × 10^5^	8.7 × 10^4^	5.5 × 10^4^
7	0	0	0
7A	0	0	0
8	4.2 × 10^5^	4.3 × 10^4^	8 × 10^2^
8A	2.1 × 10^5^	2.0 × 10^4^	2.0 × 10^4^

^a^ Sheep blood agar is an enriched, differential (hemolysis) bacteriological medium used to permit the broadest range of bacteria to grow. ^b^ MacConkey agar is a selective (for Gram negative bacteria) differential (lactose fermentation) medium. Lactose fermentation is consistent with bacteria that are members of the order *Enterobacteriales*. ^c^ All lactose negative colonies were screened for oxidase production to validate *Salmonella* testing (oxidase negative).

**Table 3 ijerph-17-05512-t003:** Retention Times, Collision Energy, and Transition States for Kratom Alkaloids, Opiates, and Benzodiazepines.

Target	Amount (µg/mL)	MF	Exact Mass	M + H	Retention Time	CID	Transition
Kratom Alkaloids							
Mitragynine	-	C23H30N2O4	398.2206	399.2279	5.55	30	399.23 --> 174.09
7-Hydroxymitragynine	-	C23H30N2O5	414.2155	415.2228	2.58	30	415.22 --> 190.09
Opioids							
Buprenorphine	100	C29H41NO4	467.3036	468.3109	3.83	45	468.3 --> 396.2
Codeine	100	C18H21NO3	299.1521	300.1594	1.37	56	300.2 --> 165.0
Fentanyl	10	C22H28N2O	336.2202	337.2275	3.74	44	337.5 --> 188.0
Hydrocodone	100	C18H21NO3	299.1521	300.1594	2.11	32	300.2 --> 199.2
Hydromorphone	100	C17H19NO3	285.1365	286.1438	0.80	36	286.2 --> 185.1
Meperidine	100	C15H21NO2	247.1572	248.1645	3.01	37	248.2 --> 220.0
(±)-Methadone	100	C21H27NO	309.2093	310.2166	4.31	43	310.2 --> 105.0
Morphine	100	C17H19NO3	285.1365	286.1438	0.63	44	286.2 --> 165.2
Naloxone	100	C19H21NO4	327.1471	328.1544	1.27	35	328.3 --> 212.1
Naltrexone	100	C20H23NO4	341.1627	342.1700	1.87	40	342.3 --> 212.2
Oxycodone	100	C18H21NO4	315.1471	316.1544	1.91	20	316.2 --> 298.1
Oxymorphone	100	C17H19NO4	301.1314	302.1387	0.69	24	302.1 --> 284.2
cis-Tramadol HCl	100	C16H25NO2	263.1885	264.1958	2.71	31	262.2 --> 58.1
Benzodiazepines							
Alprazolam	250	C17H13ClN4	308.0829	309.0902	6.83	25	309.1 --> 281.1
						40	309.1 --> 205.1
Clonazepam	250	C15H10ClN3O3	315.0411	316.0484	6.94	25	316.1 --> 270.1
						40	316.1 --> 214.1
Diazepam	250	C16H13ClN2O	284.0716	285.0789	8.28	25	285.1 --> 154.0
						30	285.1 --> 193.1
Flunitrazepam	250	C16H12FN3O3	313.0863	314.0936	7.49	25	314.1 --> 268.1
						35	314.1 --> 239.2
Lorazepam	250	C15H10Cl2N2O2	320.0119	321.0192	6.88	20	321.0 --> 275.1
						33	321.0 --> 229.1
Nitrazepam	250	C15H11N3O3	281.0800	282.0873	6.33	25	282.1 --> 236.2
						35	282.1 --> 180.1
Oxazepam	250	C15H11ClN2O2	286.0509	287.0582	6.54	20	287.0 --> 241.2
						35	287.0 --> 104.0
Temazepam	250	C16H13ClN2O2	300.0666	301.0739	7.64	20	301.0 --> 255.1
						40	301.0 --> 177.1

**Table 4 ijerph-17-05512-t004:** Levels of Mitragynine in Kratom Products.

Product #	Mitragynine (mg/g)	Opioid Screen	Benzodiazepine Screen
1	3.99	ND	ND
1A	3.35	ND	ND
2	11.33	ND	ND
2A	8.03	ND	ND
3	10.13	ND	ND
3A	9.99	ND	ND
4	4.55	ND	ND
4A	9.25	ND	ND
5	8.16	ND	ND
5A	10.05	ND	ND
6	7.54	ND	ND
6A	8.00	ND	ND
7	59.76	ND	ND
7A	60.36	ND	ND
8	9.34	ND	ND
8A	10.12	ND	ND

Two separate samples of each product were analyzed for levels of the mitragynine as described in the Methods section. Numerical values indicate mg of mitragynine per g of raw product. Note that the values for the replicate samples are generally in excellent agreement with each other. Sample #7 was a concentrated kratom extract. ND: not detected.

**Table 5 ijerph-17-05512-t005:** Levels of Metals in Kratom Products.

Product #	Metal Concentration (μg/g Raw Kratom Product)						
	As	Cd	Cr	Fe	Hg	Ni	Pb
1	0.24	0.029	3.9	348	0.018	2.4	0.32
1A	0.24	0.043	2.6	420	0.012	2.8	0.40
2	0.09	0.028	1.6	187	0.009	1.0	0.16
2A	0.22	0.024	2.2	460	0.013	3.0	0.53
3	0.20	0.028	3.5	371	0.013	2.6	0.34
3A	0.36	0.025	5.7	710	0.014	4.3	0.45
4	0.12	0.028	0.21	259	0.017	1.4	1.6
4A	0.18	0.063	4.3	430	0.012	7.4	0.34
5	0.27	0.032	2.6	850	0.010	2.6	0.41
5A	0.29	0.050	4.8	640	0.025	3.8	0.025
6	0.25	0.040	1.5	542	0.016	3.4	0.25
6A	0.19	0.027	2.5	510	0.017	3.4	0.39
7	ND	ND	ND	ND	ND	ND	ND
7A	ND	ND	ND	1.6	ND	ND	ND
8	0.11	0.028	0.39	270	0.014	0.73	0.45
8A	0.12	0.020	0.86	350	0.012	1.6	0.58

Two separate samples of each product were analyzed for levels of the metals as described in the Methods section. Numerical values indicate µg of metal per g of raw product. ND indicates that the level of metal in the sample was below the level of detection. Note that the values for the replicate samples are generally in excellent agreement with each other. All of the samples except #7 which was a concentrated kratom extract contained measurable levels of metals. As would be expected for leaf-based products, levels of iron were consistently high. Of the toxic metals that were evaluated, the highest levels were for Pb, Ni and Cr.

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
