# Peer review of "Evaluation of the Mitragynine Content, Levels of Toxic Metals and the Presence of Microbes in Kratom Products Purchased in the Western Suburbs of Chicago"

_ijerph, 2020, doi:10.3390/ijerph17155512_

Round 1

Reviewer 1 Report

The manuscript intitled “Evaluation of the Mitragynine Content, Levels of Toxic Metals and the Presence of Microbes in Kratom Products Purchased in the Western Suburbs of Chicago.” is a very interesting manuscript, excellent written and well conducted.

My main concern about the daily intake of metals by Kratom consumers is the possible synergic effects of all the metals present in each sample. Considering a daily intake of ~15 g of product, we have to multiply all the values by 15 to simulate daily exposure, as for example for sample 1: 3.6 µg As; 0.45 µg Cd; 58.5 µg Cr; 5 µg 220 Fe; 0.27 µg Hg; 36 µg Ni; 4.8 µg Pb. It would be very interesting to study this quantity of metals all together in these concentrations, assessing the toxicity of the metals present in the samples considering a daily intake of 15g. This could be done, for example, using in vitro cell culture. The authors could also explore if there are studies reporting synergic effects of 2 or more metals using lower concentrations.

I only have a minor revision considering Table 3: the first table present in the manuscript is table 3 and at the middle of the manuscript the authors refer the results of table 3. When the reader is looking for table 3 they could thought that table 3 is missing because after table 2 appears table 4. So, my suggestion is to namely the first table that appears in the manuscript as table 1. Then, if the reference of table 1 appears even at the middle of the manuscript the reader will be immediately look for it at the beginning of the document.

Author Response

This has been fixed. Please see the attached document.

Reviewer 2 Report

The manuscript is interesting and well written.

1. However in the microbiological part there is not necessary to describe microbiological media below the table, I think. These informations may be transfer to the text.

2. Moreover, MacConkey agar is not enough selective medium, therefore the Authors could used more appropriate media for Enterobacteriaceae detection (such as VRBG) or typical chromogenic agar for Salmonella sp.

3. The identification should be confirm with more specific methods such as MALDI Tof or PCR and sequencing. API Tests often give inconclusive results. 

Author Response

(The authors gave the same response as above.)

Reviewer 3 Report

  1. This paper is well-written and easy to read and understand
  2. My recommendation is that authors show representative chromatograms in the results section with peaks and retentions times.
  3. line 177 - put the number 5 in parenthesis after the worded form of the number.

Author Response

This has been fixed. Please see attached document.
